# Variability in Cadmium Uptake in Common Wheat under Cadmium Stress: Impact of Genetic Variation and Silicon Supplementation

**Rui Yang** [1,2] **, Xi Liang** [1,*] **and Daniel G. Strawn** [3]

1   Department of Plant Sciences, University of Idaho, Aberdeen Research and Extension Center, 1693 S 2700 W, Aberdeen, ID 83210, USA; yangrui01@caas.cn
2   Institute of Urban Agriculture, Chinese Academy of Agricultural Sciences, Chengdu 610213, China
3   Department of Soil and Water Systems, University of Idaho, 875 Perimeter Drive MS 2335, Moscow, ID 83844, USA; dgstrawn@uidaho.edu
*   Correspondence: xliang@uidaho.edu

**Abstract:** To decrease the transfer of cadmium (Cd) to the food chain, it is essential to select wheat (*Triticum aestivum* L.) germplasm that accumulates the least amount of Cd and to develop management practices that promote a reduction in Cd uptake. This requires knowledge of factors controlling Cd accumulation in wheat plants, which are not fully understood. The aim of this study was thus to investigate variations in Cd accumulation, translocation, and subcellular distribution in response to Cd stress and supplemental Si in two wheat cultivars that have high vs. low Cd accumulation capacities. Cd uptake and distribution in two common wheat cultivars, high-Cd 'LCS Star' and low-Cd 'UI Platinum' were evaluated at two levels of Cd (0 and 50 μM) and Si (0 and 1.5 mM) in a hydroponic experiment. LCS Star and UI Platinum were not different in root Cd accumulation but differed in Cd concentration in the shoot, which agreed with the variation between the two cultivars in their subcellular Cd distributions in organelle and soluble fractions as well as induced glutathione synthesis in response to Cd addition. Supplemental Si reduced Cd uptake and accumulation and suppressed Cd-induced glutathione synthesis. The differences between the wheat cultivars in Cd accumulation in shoots mainly derive from root-to-shoot translocation, which is related to subcellular Cd distribution and Cd-induced glutathione synthesis. Exogenous Si could decrease Cd translocation from root to shoot to alleviate Cd toxicity in common wheat.

**Keywords:** apoplastic bypass flow; Cd subcellular distribution; glutathione; root-to-shoot translocation

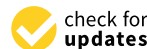



## 1. Introduction

Cadmium (Cd) is one of the most toxic contaminants and is a risk to the environment and the quality of our food [1]. Cd can accumulate in cereal grains (e.g., rice (*Oryza sativa* L.), wheat (*Triticum aestivum* L.), and barley (*Hordeum vulgare* L.)) and transfer to the food chain via the consumption of cereal-derived food products [2,3]. Therefore, developing cereal germplasm with low Cd accumulation capacities could be a promising approach to reducing Cd transfer to the food chain and subsequent Cd toxicity. This approach requires an improved understanding of the mechanisms controlling Cd accumulation and Cd tolerance in cereal plants.

After being absorbed into the root epidermis, Cd is transported through the cortex and endodermis and then enters the stele of roots for long-distance transport to the shoot [4–7]. Root-to-shoot translocation has been identified as the major process determining Cd accumulation [6], and there are genetic differences in root-to-shoot translocation. For instance, transgenic *Arabidopsis* enhanced root-to-shoot Cd translocation and Cd accumulation in shoots compared with the wild type [8]. High-Cd genotypes had a greater percentage of Cd translocation from root to shoot (or translocation factor) in hot pepper (*Capsicum annuum*

L.) [9] and durum wheat (*Triticum turgidum* ssp. *durum*) [10,11]. Cd accumulated in the shoot can be redistributed to grain during grain development, which plays an important role in Cd accumulation in grain [7,9].

After entering plant cells, Cd can be sequestered in the cell wall by binding to phosphates, proteins, peptides, and polysaccharides [12–14]. For detoxification and tolerance of the stress, Cd can also be transported to the vacuole as free Cd or Cd-phytochelatin complexes [15,16]. Phytochelatins are usually synthesized from glutathione in response to heavy metals such as Cd [17,18]. Glutathione also acts as a scavenger of reactive oxygen species to alleviate the redox imbalance caused by Cd toxicity [12,19,20]. Genetic variations have been found in Cd subcellular distributions (e.g., cell wall, organelles, and cytosol) and syntheses of glutathione and phytochelatins. In common wheat, more Cd may be sequestered in the cell wall than in the organelles and cytosol, but genotypes of different sensitivities to Cd stress may differ in Cd concentration in individual subcellular fractions [12,21]. A Cd-tolerant genotype of common wheat may have a greater glutathione concentration compared with the Cd-sensitive genotype [12], whereas the wild-type *Arabidopsis* had smaller concentrations of glutathione and phytochelatins and a lower Cd accumulation in shoots compared with transgenic plants with an enhanced ability to synthesize glutathione [18].

Exogeneous Si has shown ameliorative effects on Cd-stressed plants via the modulation of multiple processes and pathways in Cd uptake and accumulation. Si addition can decrease Cd uptake by an increase in root exudates (e.g., oxalate) that bind to Cd and prevent its absorption into the plant root, thus reducing Cd bioavailability [22]. Greger and Landberg [23] showed that Cd accumulation in wheat grain was reduced in response to supplemental Si due to a decrease in the translocation of Cd from root to shoot. Si addition also decreased Cd accumulation in subcellular fractions (e.g., cell wall, organelles, and cytosol) in the shoots of rice [24] and peanut (*Arachis hypogaea* L.) [25], and roots of common wheat [22]. Cd-induced oxidative damage (e.g., membrane lipid peroxidation and reactive oxygen species) can be alleviated by supplemental Si because Si regulates the ascorbate–glutathione cycle and improves antioxidant defense against Cd toxicity [17,20,26]. Exogeneous Si can regulate phytochelatin synthesis by regulating the phytochelatin synthase gene [17,27].

The plant accumulation of Cd is highly variable due to differences in uptake, translocation, and tolerance capacity among different plant species and genotypes. Many studies have shown contradictory Cd bioaccumulation, distribution, and toxicity effects, which likely is a result of diversities in strategies and mechanisms to tolerate Cd toxicity [18,19,21,22,27]. Despite plenty of evidence on exogenous Si improving the plant tolerance of Cd stress [22,27–29], there is a lack of investigations of the alleviatory role of Si in diverse genetic backgrounds of common wheat, especially comparing cultivars of high vs. low uptakes of Cd. In our previous experiments, we identified wheat cultivars of low and high Cd accumulation in grain and its correlation with root morphological characteristics [30], but we were unable to investigate the Cd translocation from root to shoot or the Cd subcellular distribution in field studies. We hypothesized that differences in Cd uptake and accumulation under Cd stress between common wheat genotypes can be diminished by Si amendments. Investigating plant physiological processes under Cd stress allows for insights into plant responses that would otherwise be difficult to observe. Such information allows for the comparison of processes controlling Cd uptake in high vs. low Cd uptake wheat cultivars. Thus, the objective of the current study was to investigate variations in Cd accumulation, translocation, and subcellular distribution in response to Cd stress and supplemental Si in wheat cultivars of high and low Cd accumulation capacities. This information is essential for selecting common wheat germplasms with a low Cd accumulation and developing management practices to reduce Cd accumulation in common wheat.

## 2. Materials and Methods

### 2.1. Plant Materials and Experimental Design

A greenhouse experiment was conducted in a hydroponic system at the Aberdeen Research and Extension Center, the University of Idaho in Aberdeen, Idaho (40.95° N, 112.83° W; elevation 1342 m). Wheat seeds were surface sterilized in 0.5% NaOCl solution for 20 min and placed on filter paper saturated with deionized water in Petri dishes for five days. Four seedlings of similar sizes were selected and transferred to a 2.5 L plastic pot containing modified Hoagland solution with 5 mM $Ca(NO_3)_2$, 5 mM $KNO_3$, 1 mM $KH_2PO_4$, 2 mM $MgSO_4$, 45 µM $H_3BO_3$, 10 µM $MnCl_2$, 20 µM EDTA-Fe, 0.8 µM $ZnSO_4$, 0.3 µM $CuSO_4$, and 0.4 µM $Na_2MoO_4$. The pH of the nutrient solution was measured weekly and adjusted to $6.0 \pm 0.2$ by adding 1 M HCl. The acidic environment reflects a common soil pH in many wheat growing regions that promotes high Cd bioavailability.

The experiment followed a randomized complete block design with four replications and was repeated twice in 2018 (from February 19 to March 21 and from April 9 to May 9). Two levels of Si (0 and 1.5 mM) and Cd (0 and 50 µM) were applied to two spring wheat cultivars: low-Cd 'UI Platinum' and high-Cd 'LCS Star' [30]. The level of Cd addition was chosen to create toxicity in plants in order to test plant biophysical and biochemical processes under Cd stress, and the level of Si addition was chosen to create significant impacts on Cd uptake in plants subject to Cd stress [22,24,27]. The Si and Cd treatments were initiated seven days after seedings were transferred to the hydroponic pots and supplied as $Na_2SiO_3$ and $CdCl_2$, respectively. An equivalent amount of Na as $Na_2SO_4$ was added to the zero Si treatment to compensate for the Na content of 1.5 mM Si-treated plants. The hydroponic solution in each pot was continuously aerated using an air pump and replaced every seven days with a new solution of the corresponding treatment, to maintain the Cd and Si levels in each pot generally stable during the experiment. After exposure to the Si and Cd treatments for 21 days, plants were harvested at an early booting stage (Zadoks 39–41) [31]. Root Cd uptake capability does not vary greatly over time, and the root-to-shoot translocation of Cd can reach a steady state during vegetative stages [32]; therefore, we had a one-time harvest of the plants close to the end of vegetative growth during each repeat of the experiment. The average air temperature was 24.4 °C during the first repeat and 24.0 °C during the second repeat in the greenhouse.

### 2.2. Biomass and Root Morphology

The shoot and root of each plant were separated at harvest. Roots were soaked in ice-cold 10 mM $CaCl_2$ solution for 10 min to displace extracellular Cd, rinsed in deionized water, and dried with tissue paper. The fresh roots were scanned using a high-resolution scanner (Expression STD4800, Epson America, Inc., Long Beach, CA, USA), and the images were analyzed using the WinRHIZO Pro 2013 software (Regent Instrument Inc., Quebec, QC, Canada) for root morphological characteristics including root length, surface area, volume, and average diameter. A part of the fresh shoots and roots were frozen in liquid nitrogen and stored at −80 °C until further analysis. The rest of the shoot and root samples were dried in an oven at 75 °C until the constant weight was achieved.

### 2.3. Cd Concentration and Subcellular Distribution in Shoot and Root

Dried shoots and roots were ground to a fine powder using a cyclone sample mill (UDY Corporation, Fort Collins, CO, USA) and digested following the method of Huang and Schulte [33]. In detail, 0.5 g of plant tissue was added to 5 mL of concentrated $HNO_3$ and pre-digested at 60 °C for 30 min using a block digestor (Environmental Express, Charleston, SC, USA). Then 3 mL of 30% $H_2O_2$ was added to the warm samples. Samples were further digested at 120 °C for 90 min. The final digested solution was filtered through Whatman No. 42 filter paper (GE Healthcare, Chicago, IL, USA), diluted to 50 mL using deionized water, and analyzed for total Cd using an inductively coupled plasma (ICP) spectrometer standardized using certified Cd standards (iCAP 6500, Thermo Scientific, Waltham, MA, USA).

The translocation factor (TF) of Cd was calculated as Cd concentration in shoot divided by Cd concentration in root multiplied by 100 [25]. TF has been widely used to quantify Cd translocation from root to shoot [12,15,22,25]. Cd content in the shoot or root biomass was expressed as µg plant$^{-1}$, which resulted from multiplying Cd concentration (µg g$^{-1}$) by either shoot or root biomass (g plant$^{-1}$). Cd uptake per unit root length (µg m$^{-1}$) was calculated as the amount of Cd accumulated in the whole plant (shoot and root) divided by root length (m plant$^{-1}$).

Subcellular distribution of Cd was determined according to Weigel and Jäger [34] with slight modifications. Although operationally defined, the extraction is widely used to provide information on processes that are responsible for metal uptake, translocation, and storage, and is especially relevant for single variable comparisons, such as the high vs. low Cd uptake cultivars tested in this study. Briefly, lyophilized shoots and roots were homogenized in a fractionation solution containing 250 mM sucrose, 50 mM Tris-HCl buffer (pH 7.5), and 1 mM dithioerythritol (DTE). The homogenized solution was filtered through cheesecloth, and the residue on the cheesecloth was washed three times with the same fractionation solution. To obtain the total cell wall, the filtrate was centrifuged at 300 g for 30 s. The gained pellet plus the plant residues retained on the cheesecloth was regarded as cell wall fraction. The filtrate was then centrifuged at $20,000 \times g$ for 45 min, and the obtained pellet was considered as organelle fraction, while the supernatant was the soluble fraction. The whole procedure was accomplished at 4 °C. The concentration of Cd in the soluble fraction was determined directly using the ICP spectrometer, whereas the cell wall and organelle fractions were firstly digested using the above-described method and then determined using the ICP spectrometer.

### 2.4. Apoplastic Bypass Flow

Apoplastic bypass flow was estimated by analyzing the transport of fluorescence tracer trisodium 8-hydroxypyrene-1,3,6-trisulfonate (PTS) from root to shoot following the method of Yeo et al. [35]. Immediately prior to harvest, one seedling from each pot was transferred to the same nutrient solution with the corresponding Si and Cd levels containing 30 mg L$^{-1}$ PTS. After 24 h, 1 g of the fresh shoot was cut into small pieces and extracted in 10 mL of deionized water at 90 °C for 2 h. The fluorescence was measured at 403 nm excitation and 510 nm emission (Biotek Epoch 2, BioTek Instruments, Inc., Winooski, VT, USA).

### 2.5. Glutathione in Shoot and Root

Concentrations of glutathione and glutathione disulfide in shoot and root were determined according to Dempsey et al. [36] with slight modifications. A total of 0.2 g of frozen plant tissue was homogenized with 1 mL of 0.1 M HCl and 0.1 mM EDTA (assay buffer) and centrifuged at $20,000 \times g$ for 15 min to get a clear supernatant. Then a fresh working solution, containing the assay buffer, glutathione reductase (500 U, 20 µL mL$^{-1}$ assay buffer), and DTNB (1.5 mg mL$^{-1}$ dimethyl sulfoxide, *w/v*) in a ratio of 9.1:1:1 was prepared. For total glutathione (glutathione + glutathione disulfide), 20 µL of the supernatant was mixed thoroughly with 80 µL of the working solution and 80 µL of an NADPH solution (0.25 mg reduced NADPH mL$^{-1}$ assay buffer, *w/v*). For glutathione disulfide, 4 µL of 10% 2-vinyl pyridine in the assay buffer (*v/v*) was mixed with 20 µL of the supernatant and incubated at room temperature for 60 min. Then 13 µL of 9% triethanolamine (TEA) in the assay buffer (*v/v*) was added. After incubation at room temperature for another 10 min, 80 µL of the working solution and 80 µL of the NADPH solution were added and thoroughly mixed. The absorbance was measured at 412 nm and the concentrations of total glutathione and glutathione disulfide were expressed as glutathione equivalents. The difference between total glutathione and glutathione disulfide is presented as the glutathione concentration.

### 2.6. Statistical Analysis

Data were analyzed using the generalized linear mixed model (proc glimmix) in SAS (version 9.4, SAS Institute Inc., Cary, NC, USA). For all data analyses, data from the two repeats were combined. Fixed effects included Cd level, Si level, wheat cultivar, and their interactions, and repeat and block were treated as random effects. Since the Cd treatments were significant in most variables and had interactive effects with other factors, data analysis in all variables was performed by Cd level. All variables (or variable transformations) were visually analyzed for variance homogeneity and passed the Shapiro–Wilk W test ($p > 0.05$) for normality. Treatment effects were considered significant at $p \leq 0.05$, and pairwise comparisons were made using the lsmeans statement with the Fisher's Least Significant Difference method at a significance level of 0.05. Spearman's correlation analysis was conducted using the corrplot package in R [37]. Preparation of all figures was performed using R.

## 3. Results

### 3.1. Shoot and Root Biomass and Root Morphology

Without Cd addition, UI Platinum + Si produced greater shoot biomass than UI Platinum–Si and LCS Star regardless of the Si level (Figure 1a). Under Cd stress, there was no difference in shoot biomass between the two wheat cultivars, but Si supplementation improved the shoot growth (Figure 1b). No difference in root biomass was found between the two cultivars at either Cd level (Figure 1c,d). Si supplementation improved the root growth in Cd-treated plants, but the root biomass of the -Cd + Si treatment was slightly smaller than -Cd-Si plants.

LCS Star had greater root length, surface area, and volume than UI Platinum without Cd addition (Table 1). Under Cd stress, the two cultivars showed minimal differences in root morphological characteristics, but supplemental Si increased the root length and surface area and decreased the average diameter. UI Platinum + Si and LCS Star + Si had a greater root volume than LCS Star–Si and UI Platinum–Si when the plants received Cd addition. Cd content in shoot or root was not significantly different between cultivars or Si levels at either Cd level (Figure 1e–h).

**Table 1.** Root morphological characteristics affected by cultivar and Si level without and with Cd addition.

| Treatment | Root Length m plant$^{-1}$ | Root Surface Area cm$^2$ plant$^{-1}$ | Root Volume cm$^3$ plant$^{-1}$ | Root Average Diameter mm |
|---|---|---|---|---|
| **No Cd applied** | | | | |
| Cultivar | | | | |
| UI Platinum | 24.70 B * | 234.3 B | 1.778 B | 0.301 |
| LCS Star | 28.39 A | 262.6 A | 1.953 A | 0.293 |
| Si level | | | | |
| No Si applied | 26.6 | 253.3 | 1.938 | 0.302 |
| Si applied | 26.5 | 243.6 | 1.797 | 0.291 |
| Source of variance | | | | |
| Cultivar | <0.001 | 0.001 | 0.037 | 0.243 |
| Si | 0.927 | 0.252 | 0.097 | 0.077 |
| Cultivar × Si | 0.592 | 0.538 | 0.525 | 0.833 |
| **Cd applied** | | | | |
| Cultivar | | | | |
| UI Platinum | 6.699 | 78.46 | 0.747 | 0.389 |
| LCS Star | 6.837 | 79.47 | 0.741 | 0.384 |
| Si level | | | | |
| No Si applied | 4.188 B | 54.25 B | 0.56 | 0.415 A |
| Si applied | 9.348 A | 103.7 A | 0.925 | 0.359 B |
| Source of variance | | | | |
| Cultivar | 0.770 | 0.830 | 0.881 | 0.521 |
| Si | <0.001 | <0.001 | <0.001 | <0.001 |
| Cultivar × Si | 0.130 | 0.060 | 0.028 | 0.780 |

* Means followed by different letters within a column differ significantly by cultivar or Si level ($p \leq 0.05$).

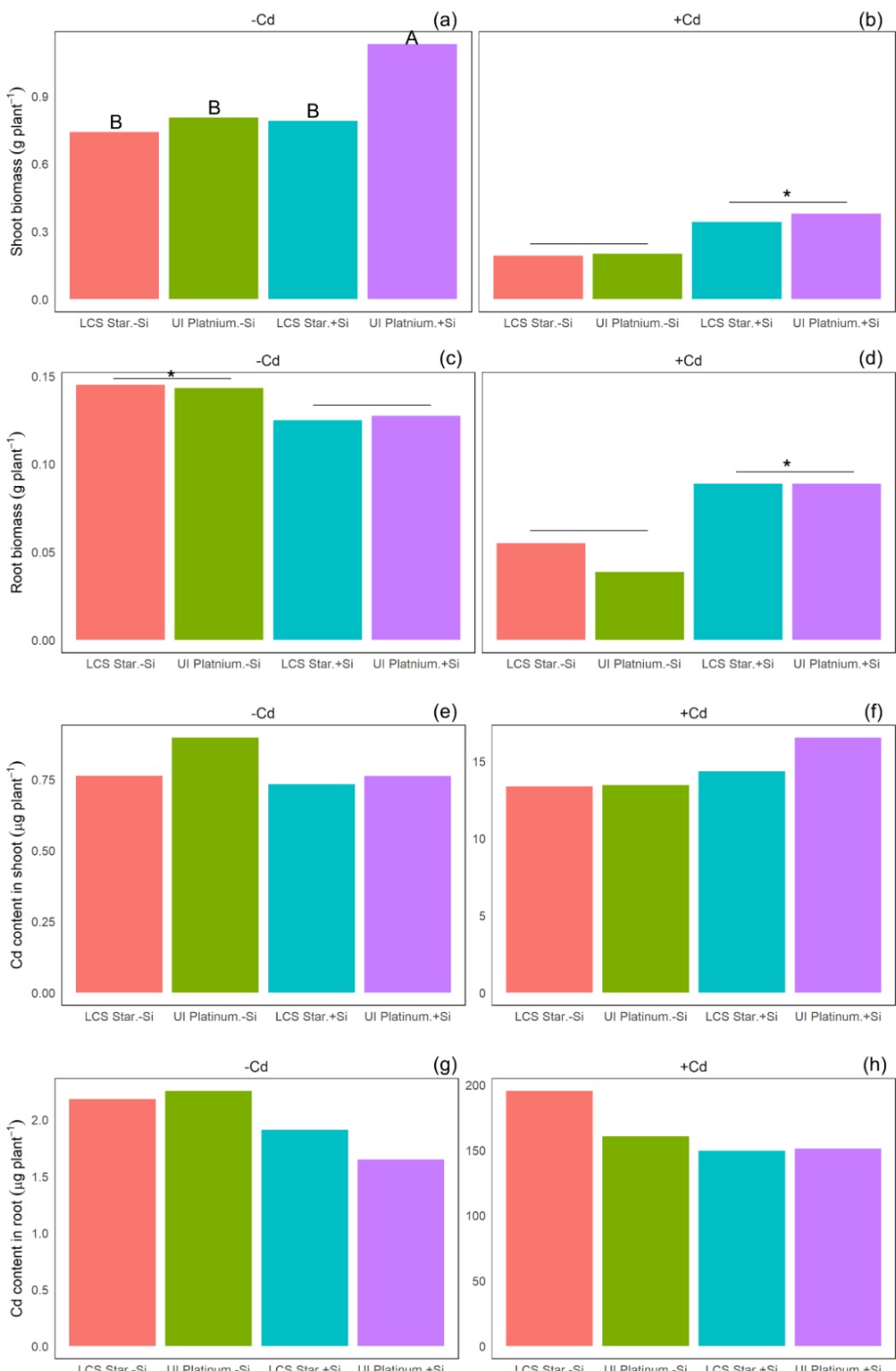

**Figure 1.** Shoot and root biomass (**a**–**d**) and Cd content (**e**–**h**) of LCS Star and UI Platinum in response to Si and Cd treatments. Cd content in shoot or root was not significantly different between wheat cultivars or Si treatments ($p > 0.05$) (**e**–**h**). Different letters indicate significant differences in cultivar × Si level at $\alpha = 0.05$. The symbol "*" indicates significant differences between Si levels at $\alpha = 0.05$.

### 3.2. Cd Concentration and Subcellular Distribution in Shoot and Root

In plants without Cd addition, the two cultivars did not differ in Cd concentration in the shoot, but supplemental Si reduced the Cd concentration in the shoot (Figure 2a). In the subcellular distribution, no difference was observed between cultivars in Cd concentration in the cell wall, organelle, or soluble fractions. Supplemental Si decreased the Cd concentration in the organelle fraction.

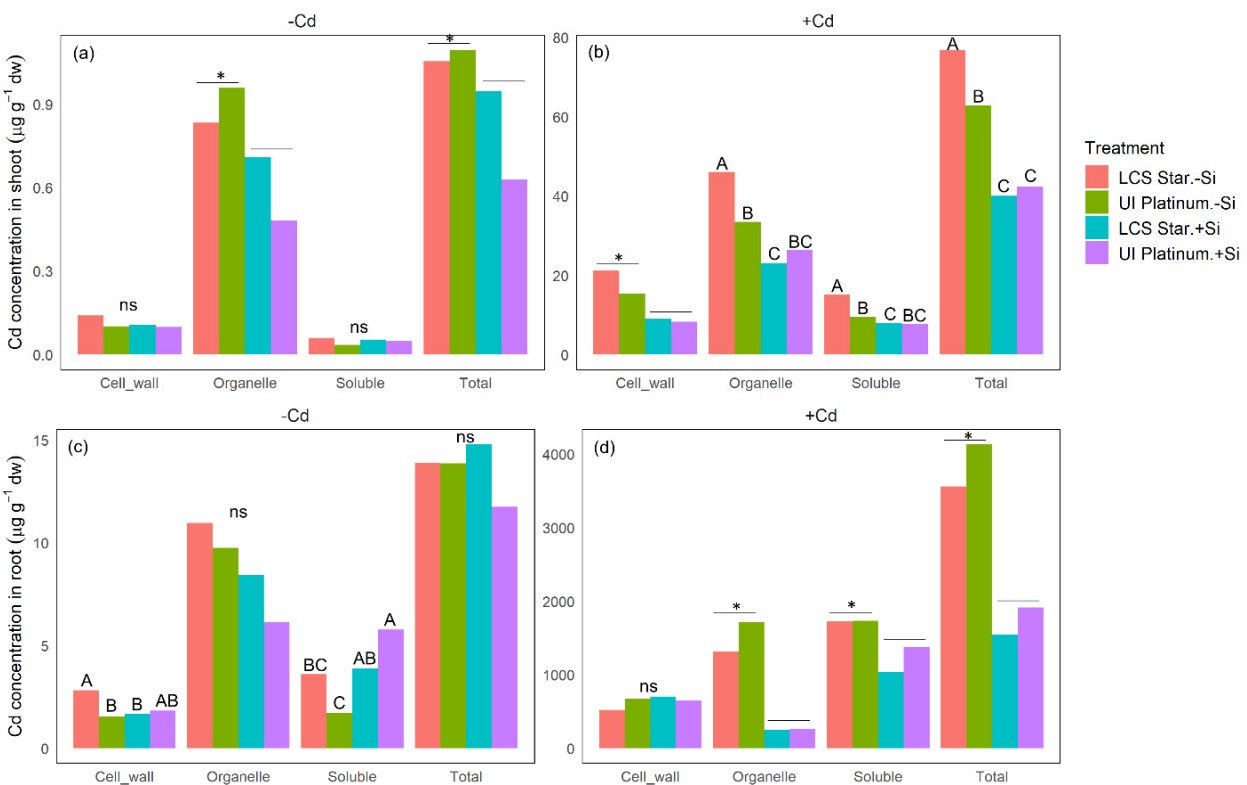

**Figure 2.** Subcellular distribution (i.e., cell wall, organelle, and soluble fractions) of Cd in the shoot (**a**,**b**) and root (**c**,**d**) of LCS Star and UI Platinum in response to Si and Cd treatments. Different letters indicate significant differences in cultivar × Si level at α = 0.05. The symbol "*" indicates significant differences between Si levels at α = 0.05. "ns" represents insignificant differences.

Under Cd stress, LCS Star–Si had the highest Cd concentration in the shoot followed by UI Platinum–Si and the two cultivars with Si supplementation (Figure 2b). Cultivars were not different in Cd concentration in the cell wall fraction, but supplemental Si reduced the concentration. Cd concentrations in the organelle and soluble fractions in LCS Star–Si was greater than in the other three treatments, but the differences between UI Platinum–Si and UI Platinum + Si in both fractions were not significant.

In the roots of plants without Cd addition, no differences were observed in Cd concentration in the whole tissue or the organelle fraction (Figure 2c). LCS Star–Si had a greater Cd concentration in the cell wall fraction than UI Platinum–Si and LCS Star + Si. A greater Cd concentration in the soluble fraction was observed in UI Platinum+Si followed by LCS Star + Si, LCS Star–Si, and UI Platinum–Si. In Cd-treated plants, cultivars were not different in Cd concentration in the root or any subcellular fraction, and supplemental Si decreased the Cd concentration in the whole root tissue and the organelle and soluble fractions (Figure 2d).

### 3.3. Glutathione (GSH) and Glutathione Disulfide (GSSG)

The concentration of GSH or GSSH in the shoot was not different between the two cultivars or Si levels in plants that were not treated with Cd (Table 2). In Cd-treated plants,

supplemental Si failed to change the concentration of GSH or GSSG in the shoot, whereas UI Platinum had smaller concentrations of both thiol products in the shoot compared with LCS Star. In the root, the concentration of GSH or GSSG did not differ between cultivars at either Cd level, and supplemental Si reduced the concentration of GSH in the root regardless of Cd level.

**Table 2.** Concentrations of glutathione (GSH) and glutathione disulfide (GSSG) in shoot and root, translocation factor (TF), Cd uptake per unit root length, and apoplastic bypass flow affected by Si level and cultivar without and with Cd addition.

| Treatment | Shoot | | Root | | TF | Cd Uptake per Unit Root Length | Apoplastic Bypass Flow |
|---|---|---|---|---|---|---|---|
| | GSH | GSSG | GSH | GSSG | | | |
| | nmol g$^{-1}$ fw | | nmol g$^{-1}$ fw | | % | µg m$^{-1}$ | nmol PTS g$^{-1}$ fw |
| **No Cd applied** | | | | | | | |
| **Cultivar** | | | | | | | |
| UI Platinum | 1046 | 527.2 | 8.391 | 1.282 | 7.174 | 0.078 | 13.34 A |
| LCS Star | 891.7 | 572.6 | 6.031 | 1.882 | 6.352 | 0.080 | 8.538 B |
| **Si level** | | | | | | | |
| No Si applied | 905.6 | 562.5 | 10.63 A | 1.406 | 8.296A | 0.092 | 12.74 A |
| Si applied | 1033 | 603.1 | 3.79 B | 1.757 | 5.494 B | 0.068 | 9.137 B |
| **Source of variance** | | | | | | | |
| Cultivar | 0.443 | 0.308 | 0.477 | 0.273 | 0.493 | 0.894 | 0.002 |
| Si | 0.546 | 0.953 | 0.050 | 0.516 | 0.026 | 0.120 | 0.015 |
| Si × Cultivar | 0.294 | 0.864 | 0.289 | 0.882 | 0.892 | 0.118 | 0.151 |
| **Cd applied** | | | | | | | |
| **Cultivar** | | | | | | | |
| UI Platinum | 916.6 B * | 583.4 B | 13.36 | 1.598 | 1.754 B | 27.85 | 2.710 |
| LCS Star | 1598 A | 756.0 A | 15.47 | 1.908 | 2.200 A | 26.59 | 3.075 |
| **Si level** | | | | | | | |
| No Si applied | 1280 | 684.4 | 18.86 A | 3.050 | 1.677 B | 44.39 A | 4.214 A |
| Si applied | 1235 | 655.0 | 9.963 B | 2.846 | 2.302 A | 16.68 B | 1.571 B |
| **Source of variance** | | | | | | | |
| Cultivar | 0.008 | 0.050 | 0.164 | 0.323 | 0.050 | 0.702 | 0.582 |
| Si | 0.850 | 0.730 | <0.001 | 0.153 | 0.010 | <0.001 | <0.001 |
| Si × Cultivar | 0.863 | 0.608 | 0.645 | 0.331 | 0.748 | 0.504 | 0.080 |

* Means followed by different letters within a column differ significantly by cultivar or Si level ($p \leq 0.05$).

*3.4. Translocation Factor, Cd Uptake, and Apoplastic Bypass Flow*

LCS Star had a greater translocation factor than UI Platinum under Cd addition; supplemental Si increased the translocation factor under Cd stress but decreased the factor without Cd addition (Table 2). Cd uptake per unit root length did not differ between cultivars or Si levels in plants that did not receive Cd addition. Under Cd stress, supplemental Si reduced the Cd uptake per unit root length, whereas no difference was found between cultivars. Si supplementation significantly reduced apoplastic bypass flow at both Cd levels. UI Platinum had greater apoplastic bypass flow than LCS Star in plants without Cd addition, while the cultivars had a similar apoplastic bypass flow under Cd stress.

*3.5. Correlations*

Without Cd addition, Cd concentrations in the shoot were positively correlated with Cd concentrations in the root and all three subcellular fractions of the shoot, and Cd uptake per unit root length (Figure 3a). Cd concentrations in the root were positively correlated with Cd concentrations in the cell wall and organelle fractions of the root and cell wall and soluble fractions of the shoot, and Cd uptake per unit root length, but negatively correlated with translocation factor.

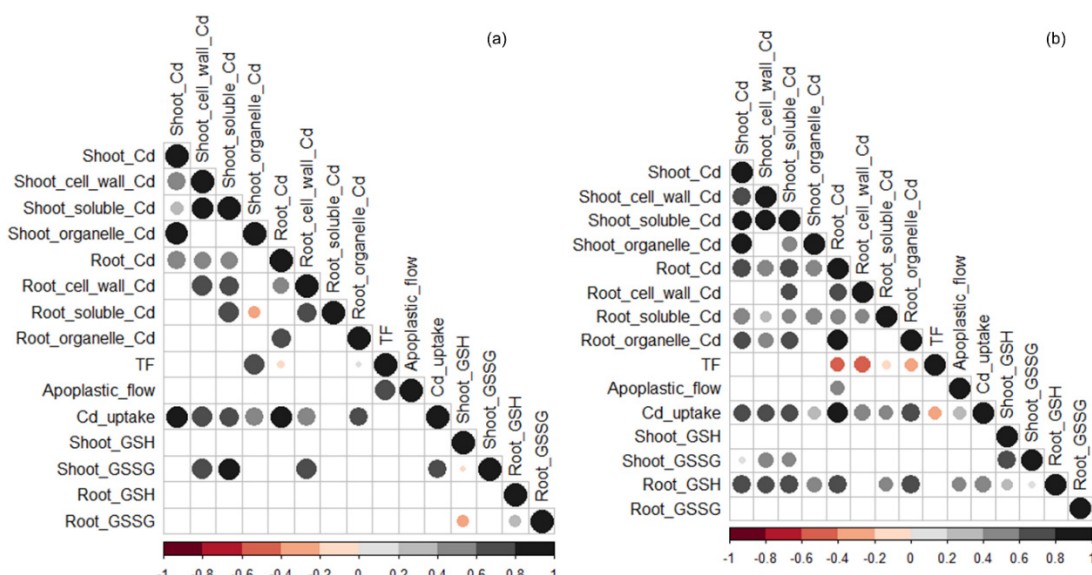

**Figure 3.** Correlations between Cd concentrations in the shoot and root and their subcellular distributions (i.e., cell wall, soluble, and organelle fractions), translocation factor (TF), apoplastic bypass flow, Cd uptake per unit root length, and concentrations of glutathione (GSH) and glutathione disulfide (GSSH) in shoot and root in Cd-untreated (**a**) and -treated (**b**) plants. The circle size and color darkness indicate the correlation coefficient, and blank cells indicate insignificant correlations ($p > 0.05$).

Under Cd stress, Cd concentrations in the shoot were positively correlated with Cd concentrations in the root, all three subcellular fractions of the shoot, and soluble and organelle factions of the root, as well as Cd uptake per unit root length, GSSG concentration in the shoot, and GSH concentration in the root (Figure 3b). Cd concentrations in the root were positively correlated with Cd concentrations in all subcellular fractions of the root and shoot, apoplastic bypass flow, Cd uptake per unit root length, and GSH concentration in the root, but negatively correlated with translocation factor.

## 4. Discussion

After being absorbed into the epidermis, Cd is transported through the cortex and endodermis and then enters the stele of roots for long-distance transport to shoot [4–7]. In the current study, the two wheat cultivars were not different in Cd concentration in the root, but a greater Cd concentration was found in the shoot of LCS Star than UI Platinum in the treatment of +Cd-Si (Figure 2b,d). LCS Star also had a greater translocation factor under Cd addition (Table 2). Thus, the differences between cultivars in Cd accumulation in the shoot are mostly derived from root to shoot translocation, and the low-Cd cultivar can restrict Cd translocation from root to shoot efficiently compared with the high-Cd cultivar [11]. The positive correlation between shoot and root Cd concentrations (Figure 3) also suggests the contribution of root-to-shoot translocation to Cd accumulation in the shoot. Our findings agree with Dong et al. [5] that tall fescue and Kentucky bluegrass had similar Cd concentrations in the root, but Kentucky bluegrass transported more Cd into its root stele and had greater leaf Cd concentration. These differences could be related to transporters at the endodermis and vascular bundles, such as natural resistance-associated macrophage proteins (NRAMP) (e.g., OsNramp5 in rice) and heavy-metal ATPase (HMA) transporters (e.g., OsHMA2 in rice and HMA4 in *Arabidopsis halleri*) [4,7,38–40].

The compartmentalization of heavy metals into tissues that are less metabolically active (e.g., cell wall and vacuole) is an important mechanism of tolerance to heavy metal toxicity [4,41]. In the current study, LCS Star and UI Platinum accumulated more shoot Cd in their organelle fraction, followed by the cell wall and soluble fraction, and cultivar differences in shoot Cd concentration under Cd addition mainly derived from differences

in Cd accumulation in organelle and soluble fraction (Figure 2b,d). However, Wu et al. [22] found that more Cd accumulated in the soluble fraction than in the organelle fraction in common wheat. In another study on common wheat, the genotype insensitive to Cd stress accumulated the least Cd in the organelle, whereas in the sensitive genotype it accumulated less in the soluble fraction; in both genotypes, Cd accumulation was greatest in the cell wall [21]. In pepper cultivars, Cd in the cell wall accounted for more subcellular Cd accumulation than the organelle and soluble fractions, regardless of their differences in Cd accumulation capacities [15]. Thus, plant species and genotypes may have various intracellular distributions and thus different mechanisms to tolerate Cd toxicity [15,21,22].

In our study, glutathione concentration in the root was positively correlated with root Cd concentrations in Cd-treated plants (Figure 3b), which agrees with findings that enhancing exogenous glutathione in the rhizosphere increases Cd accumulation in the root by forming phytochelatins and sequestering Cd in the root [18,19]. Furthermore, glutathione concentration in the root was not different between the two cultivars, but the concentration in the shoot of low-Cd UI Platinum was lower than the high-Cd LCS Star (Table 2; Figure 2b), suggesting that (1) UI Platinum might have low free Cd in the cytosol of the shoot as there was no increase in the glutathione concentration in the shoot of UI Platinum between treatments with and without Cd addition (Table 2); (2) less Cd is translocated from root to shoot and thus induces less glutathione production in the shoot of UI Platinum. The cultivar differences in our study agree with Nakamura et al. [18] that Cd concentration in the shoot was smaller in the wild-type *Arabidopsis* compared with transgenic plants with an enhanced ability to synthesize glutathione in the root. Glutathione and phytochelatins can act as long-distance carriers of Cd for translocation from root to shoot [42]. Therefore, high levels of glutathione in the root are likely to affect Cd translocation and accumulation in the shoot, e.g., the positive correlation between glutathione concentration in the root and Cd concentration in the shoot (Figure 3b).

Between the Si levels, Si addition decreased the Cd concentrations in both the shoot and root as well as Cd uptake per unit root length and apoplastic flow (Figure 2 and Table 2), which agree with previous studies on common wheat [22], durum wheat [43], rice [24,44], and cotton (*Gossypium arboreum* L.) [26]. Si deposition in the root cell wall reduces the porosity of inner root tissues, especially the endodermis [24]. The apoplastic barriers reduce the amount of Cd entering the root stele and hence the root-to-shoot translocation of Cd and eventually a decreased Cd accumulation in the shoot [45]. It also agrees with the association of a low apoplastic flow with a low Cd concentration in the root and a low Cd uptake per unit root length in Cd-treated plants (Figure 3b).

Added Si also decreases Cd concentration in the shoot and root subcellular fractions (Figure 2b,d). Si can form negatively charged complexes with hemicellulose of the cell wall that is capable of binding positively charged Cd cations and sequestering them in the cell wall [46]. Additionally, Si can limit Cd transportation from the vacuole to the cytosol, promoting vacuole compartmentalization [14]. In plants subject to Cd stress, Si also decreased the glutathione concentration in roots (Table 2), suggesting reduced concentrations of free Cd in the cytosol to induce glutathione synthesis [17]. These results agree with the Si-induced decrease of Cd concentration in the soluble fraction of roots (Figure 2d). Genetically, supplemental Si can mediate Cd-induced gene expression. For instance, Si downregulates the expressing level of genes encoding transporters (e.g., HMA2, LCT1, and Nramp5) that are involved in Cd uptake and translocation [27,46–48]. Si can also regulate genes encoding glutathione-S-transferase (e.g., GSTU1 and 6 and GSTF14), phytochelatin synthase (e.g., PCS1), and stress-associated protein (e.g., SAP1 and 14) to detoxify Cd stress [17,46,49].

Variations in Cd accumulation are controlled by multiple processes, such as transmembrane transport mediated by transporters (e.g., Nramp5, HMA2, HMA3, and LCT1) and Cd-phytochelatin complexes [4,7,18,39]. However, these transporters also regulate the translocation of other elements (e.g., manganese, iron, and zinc) bidirectionally (e.g., HMA3 for influx and Nramp3 for efflux to vacuole) [4,16,39,50,51]. Moreover, glutathione

and phytochelatins can be involved in long-distance translocation [42,51]. Even though we were unable to investigate the role of phytochelatins in Cd translocation and/or toxicity alleviation in the current study due to logistical limitations, we will try to identify which regulators (e.g., transporters and phytochelatins) are the most significant for root-to-shoot translocation of Cd in common wheat in our future work. This information will facilitate the development of common wheat germplasm with low Cd accumulation.

## 5. Conclusions

This study provides physiological insights into Cd accumulation in two wheat cultivars with contrasting uptake capabilities of Cd. The wheat cultivars LCS Star and UI Platinum were not different in Cd accumulation in the root, but LCS Star had a greater Cd concentration in the shoot. Compared with UI Platinum, LCS Star also had greater subcellular Cd distributions in organelles and soluble fractions as well as induced glutathione synthesis in response to Cd addition. These differences are highly related to differences in root-to-shoot translocation between the cultivars. Moreover, supplemental Si reduced Cd uptake and accumulation and suppressed Cd-induced glutathione synthesis, and thus could decrease Cd translocation from root to shoot to alleviate Cd toxicity in common wheat.

**Author Contributions:** Conceptualization, R.Y.; methodology, R.Y. and D.G.S.; validation, R.Y., X.L. and D.G.S.; formal analysis, R.Y. and X.L.; investigation, R.Y.; resources, X.L. and D.G.S.; data curation, R.Y. and X.L.; writing—original draft preparation, R.Y.; writing—review and editing, X.L. and D.G.S.; supervision, X.L. and D.G.S.; project administration, X.L. and D.G.S.; funding acquisition, D.G.S. and X.L. All authors have read and agreed to the published version of the manuscript.

**Funding:** Funding for this project was provided by the Idaho Wheat Commission, Ardent Mills, and Nestle.

**Institutional Review Board Statement:** Not applicable.

**Informed Consent Statement:** Not applicable.

**Data Availability Statement:** The data presented in this study are available on request from the corresponding author.

**Acknowledgments:** The authors would like to acknowledge Rosa Esquivias, Maria Gonzalez, and Alex Crump for their contributions to the experimental setup and data collection.

**Conflicts of Interest:** The funders had no role in the design of the study; in the collection, analyses, or interpretation of data; in the writing of the manuscript, or in the decision to publish the results.

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
