# Peer review of "Variability in Cadmium Uptake in Common Wheat under Cadmium Stress: Impact of Genetic Variation and Silicon Supplementation"

_agriculture, doi:10.3390/agriculture12060848_

Round 1

Reviewer 1 Report

The manuscript “variability in Cadmium Uptake in Common Wheat under Cadmium Stress: Impact of Genetic Variation and Silicon Supplementation” submitted to Agriculture MDPI. Authors investigated the effects of Si on two wheat varieties under Cd stress. The study is conducted and written well but there are some issues that I am afraid to accept it. I hope my points help authors to improve their work.

What was the reason for keeping pH on 6.0 ± 0.2? it seems too low for wheat.

Line 114, “An equivalent amount of Na as Na2SO4 was added to the zero Si treatment to compensate for the Na content of 1.5 mM Si-treated plants.” Did you check the EC of the solution?

Why did you finish the study in booting stage? That was great if you examine the translocation of Cd into the seeds because this is the point that literally we are looking for.

There is no significant effect of Si on Cd contents in root and shoot. So, can we say Si is not effective?

Table 2 is not suitable because you divided the factors. You should consider three factors in this study (Si, Cd and cultivar), and their interactions.

Analysis of variance is not suitable. Your study is randomized complete block design with four replications and repeated twice. You should combine the data and then analyze it.

There is an interesting result in Fig 2. I supposed Si treatment doesn’t have any significant effect on Cd concentrations under “-Cd” treatment. How do you explain it? If in our solution there is no Cd, how we can see difference?

Please add conclusion.

Author Response

1. What was the reason for keeping pH on 6.0 ± 0.2? it seems too low for wheat.

We added a brief explanation for the acidic environment to ensure high Cd bioavailability in the revised manuscript (line 109-110). This pH is not too low for wheat production, and we have experimented with Cd updated in wheat plants in acidic soils (pH 6.0-6.3) (Liang et al 2017).

2. Line 114, “An equivalent amount of Na as Na2SO4 was added to the zero Si treatment to compensate for the Na content of 1.5 mM Si-treated plants.” Did you check the EC of the solution?

 No, we did not check EC of the solution.

3. Why did you finish the study in booting stage? That was great if you examine the translocation of Cd into the seeds because this is the point that literally we are looking for.

Thank you for the suggestion. We terminated the experiment at the boot stage to ensure differences in plant growth in response to Cd and Si treatments. The Cd level of 50 uM is toxic, and plants would die at this Cd concentration if the experiment lasted for another few weeks. Measurements, such as apoplastic flow and glutathione from dead plants would not be comparable to alive plants. We measured Cd concentrations in both vegetative tissues and seeds in our previous work (Liang et al 2017). This hydroponic experiment focused more on root to shoot translocation, which is difficult to examine in field experiments.

4. There is no significant effect of Si on Cd contents in root and shoot. So, can we say Si is not effective?

Plant biomass with supplemental Si was greater than Cd-treated plants without supplemental Si. Cd content was not different between treatments, suggesting the impact of plant biomass was greater than Cd concentration on Cd content.

5. Table 2 is not suitable because you divided the factors. You should consider three factors in this study (Si, Cd and cultivar), and their interactions.

We analyzed the data by considering all three factors and their interactions. Since the Cd treatments were significant in most variables and had interactive effects with other factors, data analysis in all variables was performed by Cd level.

 6. Analysis of variance is not suitable. Your study is randomized complete block design with four replications and repeated twice. You should combine the data and then analyze it.

We combined the data and treated block and repeat as random effects. We specified it in the data analysis section (line 202-203).  

7. There is an interesting result in Fig 2. I supposed Si treatment doesn’t have any significant effect on Cd concentrations under “-Cd” treatment. How do you explain it? If in our solution there is no Cd, how we can see difference?

The Cd concentrations in Cd-treated plants were more than 100 times in plants without Cd treatment. So, the Cd treatment was very significant.

There was a small amount of residual Cd from some reagents for making the nutrient solution. There were also certain concentrations of Cd in the wheat seeds for establishing the plants for the experiment.

8. Please add conclusion.

Done (line 396-405).

Reviewer 2 Report

Dear authors,

The paper entitled "Variability in Cadmium Uptake in Common Wheat under Cadmium Stress: Impact of Genetic Variation and Silicon Supplementation" by Rui Yang et al. is an interesting study. The manuscript is in general well organized and well written. I have only few concerns/suggestions for the authors:

1. Given the fact that the translocator factor (TF) was calculated I would expect to find some literature about the significance of this indicator in Introduction or Materials and method.
2. Generally the TF is expressed as
the ratio of HM in plant’s edible part to that in plant root but in your study this indicator is expressed as a percent and I would like to ask why did you made this choice? Usually the interpretation of the values achieved for TF are made according to the value "1"(for more literature overview see Baldi, A.; Cecchi, S.; Grassi, C.; Zanchi, C.A.; Orlandini, S.; Napoli,M. Lead Bioaccumulation and Translocation in Herbaceous Plants Grown in Urban and Peri-Urban Soil and the Potential Human Health Risk.Agronomy 2021, 11, 2444; Baker, A.J.M. and R.R. Brooks, 1989. Terrestrial higher plants which hyper accumulate metallic elements-a review of their distribution, ecology and phytochemistry. Biorecovery, 1: 81-126; Cui S, Zhou Q, Chao L (2007) Potential hyperaccumulation of Pb, Zn, Cu and Cd in endurant plants distributed in an old smeltery, northeast China. Environ Geol 51:1043–1048;  etc).
3.The statistical significance in Figure 1 e, f, g, h – presenting Cd content in root and shoot, is missing. Please check if the values from Figure 1 h are correct (Cd content in root with Cd treatment).

4. The manuscript is generally well written but I would recommend a review in terms of editing and English spelling. For example the numbers between blankets corresponding to the cited literature are superscript and should be edited to normal format.

Author Response

1. Given the fact that the translocator factor (TF) was calculated I would expect to find some literature about the significance of this indicator in Introduction or Materials and method.

Thank you for the suggestion. We added a brief justification of translocation factor to the Materials and Methods (line 151-152).

2. Generally the TF is expressed as the ratio of HM in plant’s edible part to that in plant root but in your study this indicator is expressed as a percent and I would like to ask why did you made this choiceUsually the interpretation of the values achieved for TF are made according to the value "1"(for more literature overview see Baldi, A.; Cecchi, S.; Grassi, C.; Zanchi, C.A.; Orlandini, S.; Napoli,M. Lead Bioaccumulation and Translocation in Herbaceous Plants Grown in Urban and Peri-Urban Soil and the Potential Human Health Risk.Agronomy 2021, 11, 2444; Baker, A.J.M. and R.R. Brooks, 1989. Terrestrial higher plants which hyper accumulate metallic elements-a review of their distribution, ecology and phytochemistry. Biorecovery, 1: 81-126; Cui S, Zhou Q, Chao L (2007) Potential hyperaccumulation of Pb, Zn, Cu and Cd in endurant plants distributed in an old smeltery, northeast China. Environ Geol 51:1043–1048;  etc).

Thank you for the suggestion. In Cd-treated plants, Cd concentration in root could be 50 times the concentration in the shoot. Without multiplying by 100, the TF value would be very small. Multiplying by 100 in TF calculation has been widely used in experiments with high Cd additions.

3. The statistical significance in Figure 1 e, f, g, h – presenting Cd content in root and shoot, is missing. Please check if the values from Figure 1 h are correct (Cd content in root with Cd treatment).

There was no significant difference in Cd content between wheat cultivars or Si treatments. We added a description of the statistics to the caption of Figure 1 (line 231-232). The Cd content in the root with Cd treatment (Figure 1h) is correct.

4. The manuscript is generally well written but I would recommend a review in terms of editing and English spelling. For example the numbers between blankets corresponding to the cited literature are superscript and should be edited to normal format.

Done.

Reviewer 3 Report

I suggested  minor revisions with these two following points:   1, In wheat, does higher accumulation of Cd in the shoots mean higher Cd in the seeds? Because seeds are usually the ‘actual food’ that readers might be interested in. Therefore, the significance of Cd accumulation in shoots should be addressed or highlighted in the introduction or discussion part.

2, Further explanatory information should be added in figure 3. Dose the color of the circle mean correlation coefficient ‘r’? Then what does the size of the circle mean?

Author Response

1, In wheat, does higher accumulation of Cd in the shoots mean higher Cd in the seeds? Because seeds are usually the ‘actual food’ that readers might be interested in. Therefore, the significance of Cd accumulation in shoots should be addressed or highlighted in the introduction or discussion part.

Thank you for the suggestion. We highlighted Cd translocation to grain in the Introduction (line 48-50).

2, Further explanatory information should be added in figure 3. Dose the color of the circle mean correlation coefficient ‘r’? Then what does the size of the circle mean?

Thank you for the suggestion. We have added a description of the circle size and color in the caption of Figure 3 (311-312).

Round 2

Reviewer 1 Report

The manuscript is improved and it's acceptable in current version.